# Vascular Dysfunction Is Central to Alzheimer’s Disease Pathogenesis in APOE e4 Carriers

**DOI:** 10.3390/ijms23137106

**Published:** 2022-06-26

**Authors:** Andrew N. McCorkindale, Hamish D. Mundell, Boris Guennewig, Greg T. Sutherland

**Affiliations:** 1Charles Perkins Centre and School of Medical Sciences, Faculty of Medicine and Health, University of Sydney, Camperdown, NSW 2050, Australia; amcc2368@uni.sydney.edu.au (A.N.M.); hamish.mundell@sydney.edu.au (H.D.M.); 2Brain and Mind Centre and School of Medical Sciences, Faculty of Medicine and Health, University of Sydney, Camperdown, NSW 2050, Australia

**Keywords:** APOE4, UK Biobank, transcriptomics, plaques, tangles, cerebral amyloid angiopathy

## Abstract

Alzheimer’s disease (AD) is the most common form of dementia and the leading risk factor, after age, is possession of the apolipoprotein E epsilon 4 allele (APOE4). Approximately 50% of AD patients carry one or two copies of APOE4 but the mechanisms by which it confers risk are still unknown. APOE4 carriers are reported to demonstrate changes in brain structure, cognition, and neuropathology, but findings have been inconsistent across studies. In the present study, we used multi-modal data to characterise the effects of APOE4 on the brain, to investigate whether AD pathology manifests differently in APOE4 carriers, and to determine if AD pathomechanisms are different between carriers and non-carriers. Brain structural differences in APOE4 carriers were characterised by applying machine learning to over 2000 brain MRI measurements from 33,384 non-demented UK biobank study participants. APOE4 carriers showed brain changes consistent with vascular dysfunction, such as reduced white matter integrity in posterior brain regions. The relationship between APOE4 and AD pathology was explored among the 1260 individuals from the Religious Orders Study and Memory and Aging Project (ROSMAP). APOE4 status had a greater effect on amyloid than tau load, particularly amyloid in the posterior cortical regions. APOE status was also highly correlated with cerebral amyloid angiopathy (CAA). Bulk tissue brain transcriptomic data from ROSMAP and a similar dataset from the Mount Sinai Brain Bank showed that differentially expressed genes between the dementia and non-dementia groups were enriched for vascular-related processes (e.g., “angiogenesis”) in APOE4 carriers only. Immune-related transcripts were more strongly correlated with AD pathology in APOE4 carriers with some transcripts such as *TREM2* and positively correlated with pathology severity in APOE4 carriers, but negatively in non-carriers. Overall, cumulative evidence from the largest neuroimaging, pathology, and transcriptomic studies available suggests that vascular dysfunction is key to the development of AD in APOE4 carriers. However, further studies are required to tease out non-APOE4-specific mechanisms.

## 1. Introduction

Dementia is one of the leading causes of death worldwide and most cases (60–70%) are due to Alzheimer’s disease (AD). The main risk factors for AD are age, female sex, and possession of one or more copies of the ε4 allele of the apolipoprotein E (*APOE*) gene (*APOE4*). Although ~25% of the population are APOE4 carriers, they make up approximately 50% of sporadic AD cases [1], with the odds of developing AD being 2–3-times higher and over 10-times higher in APOE4 heterozygotes (APOE24 and APOE34) and homozygotes (APOE44), respectively [2,3,4]. It has recently been proposed that clinical AD should be subdivided into APOE4-related and non-APOE4-related sub-types [5].

In the brain, APOE is primarily expressed in astrocytes, where it is the main lipid transporter, but it is also involved in cytoskeletal modulation [6], microglial modulation [7], and amyloid-beta clearance [8]. The *APOE* gene has three major alleles at the epsilon haplotype (E2, E3, and E4). The haplotype is made up of two single nucleotide polymorphisms (SNP), 388 T > C (rs429358) and 526C > T (rs7412), that encode either arginine or cysteine at positions 112 and 158, respectively. This results in functional differences such as isoform-specific preferences for different plasma lipoproteins [6] and differences in Aβ clearance [8]. However, the precise mechanisms through which the APOE4 allele confers AD risk remains unclear, providing a major impediment to rational therapeutic strategies.

There is evidence that the brains of APOE4 carriers differ from those of non-carriers prior to AD. Magnetic resonance imaging (MRI) studies of APOE4 carriers have shown GM and WM differences in neonates through to middle-aged cohorts (well before any AD pathology), suggesting baseline differences in brain structure [9,10]. However, there is not a consistent pattern of APOE4-related changes, with some studies suggesting more hippocampal effects [11] and others suggesting more white matter (WM) effects [12,13]. The accurate characterisation of the brain differences in APOE4 carriers is an important basis for understanding why they are at increased risk of AD.

The pattern of AD pathology in APOE4 carriers and non-carriers is generally similar but there are some subtle differences. Positron emission tomography (PET) and pathology studies suggest that APOE4 is more associated with amyloid than tau [14]. However, other studies argue that faster AD progression in APOE4 carriers is due to greater tau pathology in and around the hippocampus [15]. APOE ε status is also strongly linked to cerebral amyloid angiopathy (CAA), with both APOE4 and APOE2 carriers having more CAA than APOE3 carriers [16]. CAA refers to the build-up of Aβ in cerebral blood vessels and is the leading cause of lobar haemorrhages (stroke) [17]. There are subtle differences in Aβ species between CAA and plaques, with CAA predominantly featuring Aβ_1–40_ deposition. CAA is thought to occur in 5–7% of the non-demented elderly population [18] and in most AD cases (>80%) [19]. Possession of APOE4 allele is associated with more severe CAA [20], while in AD cases, both male sex and the APOE4 allele increase CAA severity [21]. It is not clear what the relative contributions to dementia are between AD pathology and CAA but at least one study suggests that capillary CAA and associated hypoperfusion is important in contributing to dementia [22]. Clarifying the pathology differences present in APOE4 carriers is important for understanding how APOE-CAA-AD pathology interactions build on baseline differences in brain structure.

An effective means of exploring pathomechanisms is transcriptomic analysis. We previously used machine learning on transcriptomic data to identify the amyloidogenic role of lactoferrin [23] and the contribution of *ADAMTS2* and *PRTN3* to cognitive decline in AD [24]. However, the effect of APOE status was not assessed in these studies. APOE4 status is often used as a covariate or balanced across groups in analyses, but seldomly are analyses performed separately in APOE4 carriers and non-carriers. Some recent studies have performed analyses separately in APOE4 carriers and APOE2 carriers and found factors contributing to blood–brain-barrier breakdown in APOE4 carriers [25], as well as the involvement of the complement system in the protective effect of APOE2 [26].

Machine learning, or algorithmic modelling, is a powerful approach for identifying patterns within large datasets. While the focus of many machine learning studies is classification accuracy, some algorithms such as random forest also provide “feature importance” [27]. Feature importance is a measure of the contribution of each input variable (feature) to the predictive performance of the model. Feature selection algorithms find the input variables with the highest feature importance with random forest-based feature selection methods shown to be robust and powerful, particularly the Boruta method [28,29,30,31]. Here, machine learning was used in conjunction with parametric analyses such as differential expression to find the variables that best differentiated between APOE4 carriers and non-carriers.

In the present study, we firstly define the brain structural differences in APOE4 carriers by applying machine learning to ~2100 brain MRI measurements from over 33,000 non-demented APOE-genotyped individuals. Second, we use the Religious Orders Study and Memory and Aging Project (ROSMAP) and Mount Sinai Brain Bank (MSBB) pathologically confirmed datasets to define the patterns of AD-type neuropathology in APOE4 carriers before using transcriptomic data from these two studies to identify molecular differences in AD pathogenesis between carriers and non-carriers. The ROSMAP AD cases have moderate AD pathology (mean Braak score = 4.1) and RNA-seq data from the prefrontal cortex which has low tau pathology. Meanwhile the MSBB AD cases have more severe AD (mean Braak score = 5.3) and RNAseq data from brain regions with more severe tau pathology such as the parahippocampal gyrus [32,33]. The transcriptomic analysis was performed in both datasets to identify the transcripts associated with both amyloid and tau pathology and potentially those involved earlier or later in AD pathogenesis.

## 2. Materials and Methods

### 2.1. Neuroimaging Analysis

The UK biobank (UKB) is a prospective cohort study of over 500,000 individuals with genetic, health, and imaging information. Genotype array data for all UK biobank participants were downloaded from UKB in PLINK format. rs429358 and rs7412 were then used as input in PLINK 1.9 [34] to generate APOE ε status. As of 14 October 2021, there were ~42,000 participants with neuroimaging data available. All available UKB neuroimaging image-derived phenotypes (IDPs) were downloaded before some participants were excluded based on missing APOE ε genotype, presence of neurodegenerative disease (defined by ICD-10 codes from [35]—listed in Appendix A), missing scaling factor fields, and missing diffusion tensor imaging (DTI) data. This resulted in a final cohort of 33,384 with T1, T2, and DTI data, resulting in a total of 2108 IDPs (listed in Appendix A). The volume measurements generated by the UKB were normalised using the “volumetric scaling from t1 head image to standard space f25000_2_0” variable. The diffusion imaging IDPs included the following measures from tract-based spatial statistics analysis of 48 white matter masks and probabilistic tractography of 27 WM tracts: mean fractional anisotropy (FA), mean L1/L2/L3 (directional diffusion metrics), mean intracellular volume fraction (ICVF), mean isotropic volume fraction (ISOVF), mean diffusivity (MD), mode (MO) and orientation dispersion index (OD).

The cohort was evenly split above and below the age of 65 (50.0% above and 50.0% below), with a higher proportion of females in the under 65s (57.0%) than the over 65 cohort (49.6%), with those over 80 being 61% male. In the cohort, 27.6% were APOE4 carriers (54.0% female), 15.3% were APOE2 carriers (53.2% female), and 94.9% were APOE3 carriers (53.3% female), with the genotype breakdown being 176 (0.5%) APOE2/2 (50.6% female), 4158 (12.5%) APOE2/3 (53.0%), 786 (2.4%) APOE2/4 (55.0% female), 19,830 (59.4%) APOE 3/3 (53.1% female), 7775 (23.3%) APOE3/4 (53.4% female), and 732 (2.2%) APOE4/4 (54.8%).

Machine learning was carried out in the R project environment using the ‘Boruta’ R package [36]. APOE 3/3 carriers were defined as controls, while the APOE4 carriers included APOE3/4 and APOE4/4 only to avoid the potential confounding effects of the APOE e2 allele in APOE2/4 carriers. Analyses were performed separately in males and females due to brain size differences and possible APOE4 × sex interactions. Analyses were also split based on age resulting in four analyses: under 65 female controls vs. under 65 female APOE4s, over 65 female controls vs. over 65 female APOE4s, under 65 male controls vs. under 65 male APOE4s, and over 65 male controls vs. over-65 male APOE4s. Boruta was run using the ‘Boruta’ R package with default values, except for ‘maxruns = 1000’ (default value is 100). All code is available at https://github.com/binfnstats/ROSMAP_RNAseq (accessed on 8 June 2022).

### 2.2. Neuropathology Analysis of ROSMAP Data

Quantitative neuropathology data and covariates were accessed following written approval from the Rush University Alzheimer’s Disease Centre (RADC). In total, 1260 ROSMAP participants had quantitative neuropathology data available but only 916 had complete data across all brain regions (all 60 input variables listed in Appendix A). As the pathology cohort was substantially smaller than the UKB cohort, all APOE4 carriers were included as APOE4 carriers (2/4, 3/4 and 4/4), while all non-carriers were included in the non-carrier group (2/2, 2/3 and 3/3) to maximise available data (APOE genotype breakdown in Table 1). Clinical diagnosis was derived from the final consensus diagnosis (no dementia = mild cognitive impairment + no cognitive impairment), while the AD pathology diagnosis was based on modified NIA-Reagan criteria with “definite AD” and “probable AD” defined as an AD pathology diagnosis, while the “No AD pathology” group comprised participants with “possible AD” or “no AD”. Analyses were carried out separately based on clinical and pathology diagnoses, as not all individuals with a pathology diagnosis of AD had dementia. Machine learning was done using Boruta as above, as well as using five algorithms (“glmnet”, “svmLinear”, “ranger”, “rpart”, “xgbTree”) from the “caret” package [37]. The machine learning input was the 916 individuals with complete data for the 60 variables of interest (Table 1).

### 2.3. ROSMAP RNA Sequencing Analysis

Processing of the ROSMAP RNA-sequencing data has been described previously [23]. Fastq files were downloaded from *Synapse* and samples processed through our STAR/RSEM pipeline [38,39]. RSEM count data were imported into R, where lowly expressed genes were filtered out with the remaining 20,558 genes and were normalised using the trimmed mean of means (TMM) method. A larger subset of 577 subjects was used compared to our previous studies, as we did not limit the cohort to subjects with complete neuropathology or cognitive data. Differential expression was performed using the same diagnostic groups as the neuropathology analysis (dementia vs. no dementia, AD pathology versus no AD pathology) with APOE4 carriers, again including APOE 24/34/44 and non-carrier and all other genotypes (22/23/33). Differential expression analysis was performed using edgeR with RNA integrity number (RIN), batch, sex, study (ROS or MAP), and age included as covariates in the differential expression model. Enrichment analysis was performed using the WebGestalt R package [40] with the Benjamin–Hochberg false discovery rate (FDR) threshold set at 0.05.

### 2.4. Correlation Analyses

Spearman correlations were computed in R between the 20,558 input transcripts and AD variables in APOE4 carriers and non-carriers. The ROSMAP neuropathology variables included: middle frontal amyloid, middle frontal tangles, average neuritic plaques, and slope of episodic memory decline. A preliminary analysis showed that >7000 transcripts were significantly correlated with middle frontal amyloid following Benjamin–Hochberg false discovery rate (FDR) correction, with 511 transcripts still significant using the more stringent Bonferroni correction. Due to the high number of transcripts correlated with disease markers, an arbitrary threshold of the top 200 genes (representing approximately the top 1%) was used for comparisons between APOE4 carriers and non-carriers.

The *Open Targets Platform* was used as a reference to identify genes that have previously been associated with AD [41]. The platform was accessed on 15 October 2021 and the 7213 genes associated with AD were downloaded. The genes were ranked by “overall Association Score” and the top 1000 were chosen (Appendix A).

### 2.5. MSBB Data

Bulk brain RNA-seq data from Brodmann area 36 was downloaded from the *Synapse* website and processed within our lab using the same STAR/RSEM pipeline as for the ROSMAP data. There were 233 individuals with RNA-seq data from Brodmann area 36 following outlier removal, but this was reduced to 136 following the exclusion of those without APOE ε genotyping. Due to the limited number of APOE4 carriers with NCI (*n* = 6) and MCI (*n* = 6), the two groups were again combined into a “No dementia” group, while an AD pathology diagnosis was defined based on a binarised CERAD score (Definite or Probable AD = AD pathology, possible or No AD = No AD pathology) (group sizes in Table 2). There was no significant difference in age between the APOE4 and non-carrier groups in the full cohort (APOE4 mean age 82.2, non-carrier mean age = 83.2, *p* = 0.48) or within any diagnostic group. The APOE ε genotype breakdown was similar to ROSMAP and is shown in Table 3.

In total, 23,085 genes passed the filtering threshold and were used for the differential expression and correlation analyses. Differential expression analysis was performed using edgeR based on clinical diagnosis and AD pathology diagnosis, with age, sex, RIN, and PMI included as co-variates (batch was not included due to the uneven distribution of groups across batches, but there was no clear batch effect with principal components analysis (data not shown)). The correlation analysis was performed using the same methodology as for the ROSMAP data with mean neuritic plaques, clinical dementia rating (CDR), and Braak stage used as disease markers. A random sample of 34 controls was taken from the non-carrier controls to match the AD composition of the APOE4 group (34 controls and 48 AD = 41.5%) for the correlation analysis.

## 3. Results

### 3.1. APOE4 and Neuroimaging in UKB Participants with No Dementia

There were 33,384 non-demented individuals with complete data for the 2108 image-derived phenotypes (IDPs). The APOE 3/3 group (*n* = 19,803; 53.1% female), was used as the control group, while the APOE4 group comprised APOE 4/4 (*n* = 732; 54.8% female) and APOE 3/4 heterozygotes (*n* = 7775: 53.4% female). The Boruta feature selection algorithm was applied to find the IDPs that best differentiated between APOE4 carriers and non-carriers. Analyses were performed separately based on age and sex as follows: under 65 female controls vs. female APOE4, over 65 female controls vs. female APOE4, under 65 male controls vs. male APOE4 and over 65 male controls vs. male APOE4. There was no significant difference in age or intracranial volume between APOE4 carriers and non-carriers for any analysis. In the under 65s, there were 17 IDPs different in the female analysis and 14 in the male analysis (Top ten for both in Table 4).

In the over 65s, there were 32 IDPs different between females and 44 between males (Top ten IDPs for both in Table 5). In both under 65 analyses (males and females), most IDPs were white matter (WM)-related, while in the over 65 analyses, nine of the top ten IDPs were grey matter (GM) regions. There was limited overlap of the exact fields between the male and female analyses from each age bracket, but there was overlap of the affected brain regions. Several of the top-ranked WM IDPs were in posterior regions such as the posterior thalamic radiation and posterior corona radiata, as well as high ranking GM changes in the visual cortex.

A further Boruta analysis was performed in the same four groups to find the brain regions associated with copies of the E4 allele. WM IDPs were ranked higher in the under 65s, while GM IDPs were higher in the over 65s (Appendix A). In the under 65 analyses, the top-ranked WM and GM IDPs changes were in posterior brain regions. There were no highly ranked hippocampal IDPs. In the over 65 analyses, there were multiple hippocampal IDPs in the Top 10 in the female analysis only.

To explore directionality, a correlation analysis was carried out between each IDP and copies of the E4 allele. For WM, ICVF and FA measures were generally negatively correlated with copies of the E4 allele, while ISOVF, L2, L3, and MD measures were positively correlated. GM volume of the intracalcarine cortex and altered posterior thalamic radiation integrity (ICVF and MD) were significantly positively correlated with copies of the E4 allele in both males and females after Bonferroni correction. Hippocampal volume measures were significantly negatively correlated with number of E4 alleles in the over 65 females but not in over 65 males. In over the 65 males, the number of E4 alleles was positively correlated with multiple volume measures from the parietal and occipital lobes (Appendix A).

### 3.2. The Effect of APOE4 on AD-Type Neuropathology

A total of 1260 individuals from the ROSMAP cohort with neuropathology data were used to compare the pattern of neuropathology between APOE4 carriers and non-carriers. A subset of 916 had complete neuropathology data (tau and amyloid quantified in eight different brain regions), including 234 APOE4 carriers. APOE4 carriers were more likely to have an AD pathology or dementia diagnosis (Table 1—See methods). The average amount of amyloid and tau in the ROSMAP cohort within each APOE genotype is shown in (Figure 1), with both amyloid and tangles higher in APOE4 carriers. There were larger differences in amyloid between the APOE ε genotypes than tau (Figure 1) for both total brain and individual brain regions. Neocortical areas had the highest fold change differences for amyloid and tau between APOE4 carriers and non-carriers (angular gyrus, middle frontal gyrus, and calcarine cortex highest), while hippocampal and entorhinal fold changes were lower in both AD and controls (Appendix A).

The neocortical differences in pathology between APOE4 carriers and non-carriers were further investigated using machine learning. Classification analyses were performed based on clinical diagnosis (dementia vs. no dementia (combined mild impairment and no impairment)) and AD pathology diagnosis (AD pathology vs. no AD pathology diagnosis). Five machine learning algorithms were run on the ROSMAP neuropathology data to classify participants within each diagnostic group: AD APOE4 vs. AD non-carrier, CTRL APOE4 vs. CTRL non-carrier (No AD pathology diagnosis), dementia APOE4 vs. dementia non-carrier, and no-dementia E4 vs. no-dementia non-carrier (no dementia diagnosis). The area under the receiver operatic characteristic (AUROC) curve values was highest in the dementia analysis (~0.76) and lowest in the control analyses (~0.6). The highest ranked variables in these models were mostly amyloid related with cerebral amyloid angiopathy consistently ranked highly, as well as amyloid in the calcarine cortex (Appendix A).

### 3.3. Molecular Features of the AD Brain

Bulk tissue RNA-seq data from the PFC of ROSMAP study participants were interrogated to find the molecular differences associated with APOE. Differential expression analysis (DEA) was firstly performed within diagnostic groups between APOE4 carriers (2/4, 3/4 and 4/4) and non-carriers (e.g., APOE4 carriers with AD pathology vs. non-carriers with AD pathology) in the ROSMAP-PFC dataset. Group sizes are shown in Table 6. There were few differentially expressed genes (DEG)s in any of these analyses (maximum of 12). Relaxing the FDR threshold to 0.1, sequentially removing covariates from the differential expression model, and only including APOE 3/3 as controls versus only APOE 3/4 and 4/4 as APOE4 carriers did not substantially increase the number of DEGs (All DEGs listed in Appendix A).

Due to the disconnect between the pathological diagnosis of AD and clinical diagnosis of dementia, differential expression analyses were performed separately based on clinical and pathological diagnoses (Table 6). There were 3018 (1440 upregulated) DEGs between non-carriers with (pathological) AD and non-carriers without an AD pathological diagnosis. The enrichment results showed both gene ontology (GO) terms and Kyoto Encyclopaedia of Genes and Genomes (KEGG) pathways related to metabolic processes and synaptic activity. There were only five DEGs (*ENSG00000259976*, *LRP2BP*, *ENSG00000258768*, *GPALPP1*, *UPP1*) in the corresponding APOE4 analysis. The analysis was repeated based on clinical dementia status, as 58 APOE4 carriers had no dementia compared to only 29 who did not have an AD pathology diagnosis. Within the APOE4 carriers, there were 601 DEGs (418 upregulated) between the dementia and non-dementia groups. The DEGs were enriched for GO biological process (BP) terms related to vascular development and cell migration, as well as for cytoskeletal-related processes/pathways in the GO cellular component (CC), GO molecular function (MF) and KEGG analyses. Separate enrichment analyses of the upregulated and downregulated genes showed that the up-regulated transcripts showed the same enrichment pattern, while the downregulated transcripts showed limited enrichment (mostly ribosomal/RNA binding terms) across all analyses (Complete results in Appendix A).

Within the non-carriers, there were 585 DEGs (377 upregulated) between the dementia and non-dementia groups. There was enrichment for cytoskeletal/adhesion-related terms in the GO MF, GO CC, and KEGG analyses but no GO-BP enrichment. The upregulated transcripts were enriched for cytoskeletal GO terms and the “Focal adhesion” KEGG pathway, while the downregulated transcripts showed little enrichment. The APOE4 and non-carrier DEGs were then compared with 158 (~27) overlapping DEGs. There was no GO-BP enrichment of the overlapping DEGs, while the DEGs unique to APOE4s were enriched for vascular/cell migration GO-BPs, with no enrichment of non-carrier DEGs (Table 7).

A transcript association analysis was then performed to find if different transcripts were correlated with specific disease indices within APOE4 carriers. Due to the low percentage of APOE4 carriers without AD pathology (19.5%), a random sample of the non-carrier controls was taken to balance AD composition between the groups.

A Spearman correlation analysis was performed between the gene expression data and markers of AD (middle frontal amyloid, middle frontal tangles, average neuritic plaques, and slope of episodic memory decline). There were 36 transcripts common (APOE4 carriers and non-carriers) to the top 200 for amyloid, including *CD2AP*, while *RABEP1* was higher in APOE4 carriers and *ABCA1* was higher in non-carriers; 35 common for episodic memory with higher correlations for *LTF*, *ADAMTS2*, *BACE2*, and *APOB* in APOE4 carriers; 25 common for neuritic plaques including *LTF* and *ADAMTS2* with several metallothionein (*MT1F*, *MT1E*, *MT1X*, *MT2A*) present for APOE4 carriers; and 26 common for tangles including *LTF* with enrichment for metal ion-related GO terms for the APOE4 genes, but not the non-carrier transcripts again. *C4A* and *C4B* were more strongly correlated with neuritic plaques in APOE4 carriers than non-carriers.

### 3.4. Mount Sinai Brain Bank Data

A second bulk tissue RNA-seq dataset from the parahippocampal gyrus (Brodmann area 36 (BA36)) of MSBB donors (mostly Braak stage V/VI) was used to investigate the effects of APOE4 status in a cohort and brain region with more advanced AD. A total of 136/233 subjects had APOE genotyping information and RNA-seq data from BA36 (Table 3). The differential expression and correlation analyses were repeated as described previously.

In contrast to ROSMAP, there was limited differential expression between the dementia and non-dementia groups in the APOE4 analysis, with only 18 DEGs compared to 152 based on AD pathology status (only 12 APOE4 carriers in the non-dementia group). There was no enrichment of the dementia versus non-dementia DEGs, so APOE4 versus non-carrier comparisons were performed based on AD pathology diagnosis. In the AD (pathology) versus control analysis, there were 1917 (1049 upregulated) DEGs between non-carrier AD and non-carrier controls compared to 152 for APOE4 AD and APOE4 controls, with 50 in common (Transthyretin (*TTR*) common to top 20).

Enrichment analysis (FDR < 0.05) of the APOE4 DEGs showed no GO BP enrichment so the FDR threshold was relaxed to 0.1, which resulted in 664 DEGs (232 common to the non-carrier DEGs). There was abundant enrichment with 53 GO BP terms in the non-redundant analysis (32 mostly synaptic and transport-related terms common to non-carrier analysis). The terms unique to the APOE4 group were mostly related to transport, while the terms unique to non-carriers included extracellular matrix, adhesion, and immune-related terms (complete results in Appendix A).

### 3.5. Correlation Analysis of Advanced AD

The transcript correlation analysis was repeated in the MSBB data, with AD status again balanced across the APOE4 and non-carrier groups using the available MSBB disease markers (mean neuritic plaques, clinical dementia rating (CDR), and Braak stage). The top 200 genes correlated with neuritic plaques in APOE4 carriers were enriched for immune-related GO terms such as “innate immune response” and included AD-related genes such as *TREM2*, *GFAP*, and *MSR1*, as well as multiple complement transcripts (*C3*, *C5*, *C1S*, *C4B*). There was no enrichment of the non-carrier transcripts and only one overlapping transcript in the respective top 200s (*SCAMP1-AS1*), but the non-carriers did include disease-related transcripts such as *APP*, *APOB*, and *ACHE*. For CDR, there were three genes common to both APOE4 and non-carriers (*WWTR1*, *VIM*, *EDA2R*) with enrichment for immune-related terms in the APOE4 transcripts (included *GFAP* and *MSR1* again) and no enrichment for the non-carrier transcripts (included *ADAM10*). A similar pattern was observed for the Braak score, with only *WWTR1* common to the top 200 lists.

There were some transcripts that showed opposite correlations with disease markers in APOE4 carriers compared to non-carriers. For example, *GFAP* and *TREM2* were positively correlated with neuritic plaques in APOE4 carriers but negatively correlated in non-carriers, while others such as *APP*, *EED*, and *CNTNAP5* were negatively correlated with disease markers in APOE4 carriers and positively correlated in non-carriers (*EED* and *CNTNAP5* also had disparate correlations in ROSMAP data). *ADAMTS2* and *LTF* (from our previous study) were more correlated with disease markers in APOE4 carriers (complete correlation results in Appendix A).

## 4. Discussion

Other than ageing, possession of the APOE4 allele is the largest risk factor for AD. Given the limited success of amyloid-modifying therapies, attention has turned to alternative therapeutic targets. Therefore, how APOE4 increases AD risk has become the most important question in AD research. The mean age at onset of AD is 80 years, meaning that APOE4 may only have subtle effects on brain function earlier in life [42] or, less likely, confer some survival advantage to cognitive fitness (antagonistic pleiotropy) [43]. Age related changes in brain structure and activity have been documented in aged non-demented individuals [42] but the UKB provides an unprecedentedly large resource for categorically defining the baseline effects of APOE4 allele possession on brain structure and function. A recent analysis of part of the UKB (12,662 participants) found increases in white matter hyperintensities, although with no APOE4 and age interactions [43]. Here, we expanded on this analysis in 33,384 non-demented participants and over 2100 IDPs.

Our initial neuroimaging analysis in APOE4 carriers showed subtle changes in white matter in under 65s and more prominent grey matter changes in over 65s. The top ranked IDPs tended to be in brain regions associated with CAA (e.g., posterior thalamic radiation and occipital grey matter volumes). Specifically, there was a direct correlation between copies of the APOE4 allele and increased GM volumes in the posterior brain. The occipital lobe is known to be the most affected by CAA [16]. CAA results from the accumulation of amyloid in small to moderate arteries, likely resulting from inefficient clearance that may itself result from loss of BBB integrity with ageing or contribute to it [44]. CAA is the leading cause of spontaneous lobar haemorrhages in the aged population [45]. A recent study also found that cerebral microbleeds were associated with increased GM and WM volume, while CAA was associated with increased cortical thickness in a meta-analysis of 2657 MRIs and 82 autopsies [46]. This effect was more pronounced in individuals with fewer cerebral microbleeds. A separate study of 659 individuals (83 with cerebral microbleeds) found that cerebral microbleeds were associated with increased WM volume [47]. As cerebral microbleeds and CAA are correlated with cognitive impairment [48], an increase in brain volume with the first cerebral microbleeds has been suggested as an early biomarker of neurodegeneration [49]. Overall, these results highlight a pattern of APOE4-related vascular dysfunction.

A similar pattern was observed in the neuropathology analysis with higher CAA and more severe pathology (particularly amyloid rather than tau) in APOE4 carriers in cortical regions such as the calcarine cortex. The differences in pathology were more pronounced in neocortical areas than in the hippocampus in subjects with and without a pathological AD diagnosis. APOE4 controls had significantly higher amyloid, but not tau, compared to non-carrier controls in all eight brain regions. The machine learning analysis showed that the degree of CAA and severity of amyloid pathology (particularly in the calcarine cortex) was important in distinguishing between APOE4 carriers and non-carriers within diagnostic groups. The relative importance of CAA and high ranking of amyloid in the parietal and occipital lobes overlapped with the neuroimaging results here. APOE4 carriers have been shown to have higher rates of CAA in multiple previous neuropathological studies [16,20,21], particularly the CAA-type 1 in which there is capillary occlusion [22] The other neuropathology results here support previous findings of a stronger effect on amyloid than tau pathology [50,51].

The molecular analysis built on these pathology results demonstrates that CAA-related vascular dysfunction contributes to cognitive impairment. The DEGs in the ROSMAP analysis between APOE4s with dementia and APOE4s with no dementia were enriched for several vascular-related GO terms, while the non-E4s DEGs were not. This included *ADM*, the gene encoding for adrenomedullin. *ADM* is largely expressed by cerebral endothelial cells and is known to regulate the blood–brain barrier (BBB) [52]. This effect is mediated via the upregulation of claudin-5 [53], the most highly expressed tight junction component that has been previously described as “the gatekeeper of neurological function” [54]. Aβ can downregulate claudin-5, upregulating its own permeability across the BBB [55]. This opens the possibility that ADM and claudin-5 could be therapeutically manipulated to promote Aβ clearance.

There were also differences in the transcripts correlated with disease markers between APOE4s and non-carriers, including for AD-related transcripts such as *CD2AP* and *ABCA1*. These association analyses suggest that AD pathology and cognitive impairment develop via different mechanisms in APOE4 carriers compared to non-carriers. The lack of differential expression between APOE4 AD and APOE4 controls may have been due to the elevated amyloid pathology in the controls, the relatively small size of the APOE4 control group, and the lack of tau pathology in the region where the RNA-seq samples were taken from.

In the MSBB analysis, there was limited differential expression between APOE4 dementia and APOE4 without dementia compared to 152 DEGs between APOE4 AD and APOE4 controls. This is likely related to the small number of APOE4 carriers without dementia (six with no impairment and six with mild impairment). There were considerable differences in the transcripts correlated with disease indices with several known AD genes (e.g., *APP*, *EED*, *CNTNAP5*, *TREM2*, *GFAP*) positively correlated with disease markers in one group and negatively correlated in the other (*EED* and *CNTNAP5* had disparate correlations in the ROSMAP data as well). *TREM2* was positively correlated with neuritic plaques in APOE4 carriers (Spearman correlation = 0.48) but negatively correlated with neuritic plaques in non-carriers (Spearman correlation = −0.26). This could mean that immune dysregulation has a greater role in AD development in APOE4 carriers, as the transcripts mostly correlating with neuritic plaques showed immune-related enrichment. Overall, the MSBB analysis supports the ROSMAP results of significant differences between APOE4 carriers and non-carriers in disease development. However, the greater overlap in enrichment between the APOE4 and non-carrier AD analyses in the MSBB analysis suggests that although disease mechanisms differ, the end-stage of AD where tau pathology is abundant is similar.

Transcripts from the complement system were more correlated with AD pathology in the APOE4 carriers than the non-carriers in both cohorts. The complement system is involved in signalling between cells such as microglia and astrocytes [56], as well as regulating the blood–brain barrier [57]. It was also recently implicated in mediating the protective effect against AD in APOE2 carriers [26]. This means that the complement system could have a key role in regulating the deleterious effect of APOE4, as well as the protective effect of APOE2.

A limitation of the molecular analysis was the limited number of APOE4 carriers without dementia or AD pathology, particularly in the MSBB cohort. As ~100 MSBB participants were missing APOE genotype information, this analysis could be repeated in future if more genotype data becomes available. Increased sample size may also permit a sex-specific analysis, as APOE-sex interactions have been observed previously [11] including this study concerning the neuroimaging analysis. The low overlap between the ROSMAP and MSBB datasets has been observed in previous studies [32,58]. This is likely due to more advanced AD and the exclusion of other pathologies [59] in the MSBB cohort compared to moderate AD and a high prevalence of mixed pathology in ROSMAP [60].

While employing non-parametric machine learning, including over 2000 IDPs, and having over 5000 subjects in each classification analysis can be determined as the strengths of our approach, our exploratory neuroimaging analysis has some limitations. The major confounders of age and sex were accounted for but others such as handedness, education, ethnicity, other neurological conditions, and co-morbidities such as hypertension were not. It is possible that further subdividing the groups into smaller age groups and applying more rigorous exclusion criteria could increase the accuracy of the results. However, the purpose of the analysis was to screen for changes in brain structure between APOE4 carriers and non-carriers and our results generally overlapped with the previous study [13].

### Future Directions

These results suggest that vascular dysfunction related to CAA contributes to the initial development of AD pathology and cognitive impairment in APOE4 carriers. The strong effect of CAA is significant, as anti-Aβ antibody treatments such as aducanumab are not recommended for individuals with CAA due to the high rates of amyloid-related imaging abnormalities (ARIA) [61]. ARIAs are the main adverse event associated with anti-Aβ antibodies and a meta-analysis of clinical trials showed that APOE4 carriers were at the highest risk [62]. Notwithstanding the APOE4-specific pathomechanisms described, persons receiving anti-Aβ antibodies should be genotyped and split into APOE4 carriers and non-carriers. Adjunctive treatments aimed at maximising cerebrovascular health could limit the adverse effect of anti-Aβ antibodies in APOE4 carriers.

## 5. Conclusions

This study provides multi-modal evidence that vascular dysfunction is significant in the development of AD in APOE4 carriers. Both CAA and parenchymal amyloid pathologies are more prominent than in non-carriers and these may result from the same pathomechanism. Genetic evidence suggests that subtle differences in immune processes, particularly involving TREM2, could be significant in the increased AD pathology observed in APOE4 carriers. Overall, this multi-modal analysis supports different molecular mechanisms in APOE4 carriers and non-carriers. While the pathomechanisms operating in non-carriers are less clear, the risk associated with APOE4 appears to be related to Aβ clearance through vasculature and TREM2-mediated microglial processes.

## Figures and Tables

**Figure 1 ijms-23-07106-f001:**
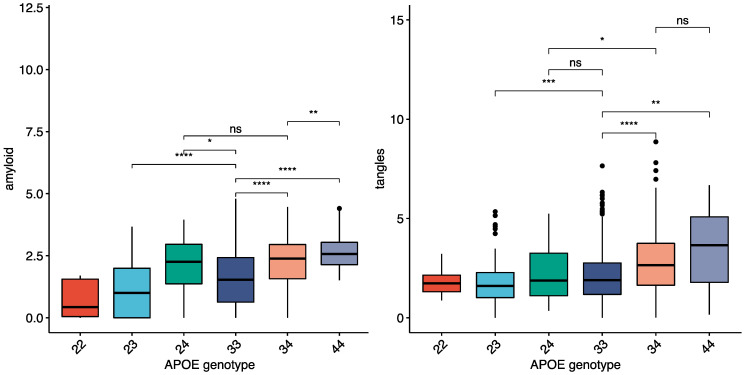
Amyloid and tau pathology in each genotype in the ROSMAP cohort. “Amyloid” is mean areal fraction of amyloid across eight brain regions (measurement available for individuals with data from at least four brain regions). “tangles” is mean density of abnormally phosphorylated tau positive cells in the same eight brain regions (measurement available for individuals with data from at least four brain regions). Ns = not significant, * = *p* < 0.05, ** = *p* < 0.01, *** = *p* < 0.001, **** = *p* < 0.0001.

**Table 1 ijms-23-07106-t001:** Total number of subjects per clinical and AD pathology diagnostic group within each APOE genotype in the 916 ROSMAP subjects used in the machine learning analysis.

	APOE 2/2	APOE 2/3	APOE 3/3	APOE 2/4	APOE 3/4	APOE 4/4
AD path	1 (16.7%)	54 (44.3%)	332 (59.9%)	17 (81%)	165 (83.3%)	13 (86.7%)
No AD path	5	68	222	4	33	2
Dementia	1 (16.7%)	40 (32.8%)	207 (37.4%)	12 (57%)	113 (57%)	11 (73.3%)
MCI	2	23	162	4	61	1
NCI	3	75	185	5	58	3
Age	93.2	89.9 *	89.8 *	89.5	89.1	84.8
Total ^	6 (0.7%)	122 (13.3%)	554 (60.5%)	21 (2.3%)	198 (21.6%)	15 (1.7%)

^ = total number and percentage of cohort with each genotype. AD path = AD pathology diagnosis according to NIA-Reagan criteria, MCI = mild cognitive impairment, NCI = no cognitive impairment. 2/3 and 3/3 group were significantly older than APOE 4/4 group (* *p* = 0.04 for both). APOE 4/4 vs. 2/2 was *p* = 0.07—limited by small group size.

**Table 2 ijms-23-07106-t002:** APOE4 status of those with clinical dementia or an AD pathology diagnosis in the MSBB cohort.

	Clinical Diagnosis		AD Pathology Diagnosis		
	No Dementia	Dementia	No AD Path	AD Path	Total
APOE4	12 (29%)	30 (71%)	15 (36%)	27 (64%)	42 (31%)
Non-carrier	27 (32%)	67 (68%)	46 (46%)	48 (54%)	94 (69%)
Total ^	39 (29%)	97 (71%)	61 (45%)	75 (55%)	136

^ = total number and percentage of cohort with each genotype. NB—Percentages are to be read horizontally showing the diagnosis composition of APOE4 carriers and non-carriers.

**Table 3 ijms-23-07106-t003:** Clinical and AD pathology diagnosis within each APOE genotype in the 136 MSBB subjects used in the transcriptomic analysis.

	APOE 2/2	APOE 2/3	APOE 3/3	APOE 2/4	APOE 3/4	APOE 4/4
AD path	0	6 (37.5%)	42 (55.3%)	1 (100%)	23 (60.5%)	3 (100%)
No AD path	2	10	34	0	15	0
Dementia	1 (50%)	7 (43.8%)	59 (77.6%)	1 (100%)	26 (68.4%)	3 (100%)
MCI	1	4	7	0	6	0
NCI	0	5	10	0	6	0
Age	88	83.1	83.3	90	82.5	76
Total	2 (1.5%)	16 (11.8%)	76 (60.5%)	1 (0.7%)	38 (21.6%)	3 (2.2%)

**Table 4 ijms-23-07106-t004:** Top 10 image-derived phenotypes for classifying the UKB participants into APOE4 carriers or APOE 3/3 controls in males and females aged under 65 years.

Under 65 Females	Under 65 Males
Mean L2—Post. thalamic radiation (L)	Mean ICVF—sagittal stratum (L)
Area—Total (R) hemisphere	Mean (w) ICVF—Inf. Long. Fasc. (L)
Mean MD—sagittal stratum (R)	Mean (w) MD—Inf. frontooccipital Fasc.(L)
Mean (w) L1—Inf. Long. Fasc. (L)	Mean ICVF—Post. corona radiata (R)
Area—V2 (L) hemisphere	Mean MD—Post. corona radiata (R)
Mean ICVF—external capsule (L)	Mean (w) L1—Sup. Long. fasciculus (L)
Mean ICVF—cingulum cingulate gyrus (R)	Volume—supratentorial not vent whole brain
Volume—V2 left hemisphere	Mean (w) L1—Inf. Long. fasciculus (L)
Mean(w) L2—Post. thalamic radiation (L)	Mean (w) L2—forceps minor
Mean ICVF—Post. limb of internal capsule (R)	Mean (w) L3—Post. thalamic radiation (L)

Mean (w) = weighted mean from probabilistic tractography, Fasc. = fasciculus, ICVF = intracellular volume fraction, Inf. = inferior, Long. = longitudinal, MD = mean diffusivity, Sup. = superior, (L)/(R) = Left/Right, L2/L3 are measures of radial diffusivity from fractional anisotropy.

**Table 5 ijms-23-07106-t005:** Top 10 image-derived phenotypes for classifying the UKB participants into APOE4 carriers or APOE 3/3 controls in males and females aged over 65 years.

Over 65 Females	Over 65 Males
Area—total surface (R) hemisphere	Area—total pial surface (L) hemisphere
Volume—supratentorial not vent whole brain	Area—total white surface (L) hemisphere
Volume—brainseg not vent surf whole brain	Area—total pial surface (R) hemisphere
Volume—basal nucleus (L)	Area—total white surface (R) hemisphere
Volume—whole hippocampus (R)	Area—Sup. frontal (L)
Volume—accessory basal nucleus (L)	Volume—Sup. frontal (L)
Volume—lateral nucleus (L)	Area—Sup. frontal gyrus (L)
Volume—whole hippocampal body (R)	Volume—Sup. parietal (L)
Area—BA1 (L)	Area—gyrus cuneus (R)
Volume—gyrus + sulcus Ant. cingulate (R)	Area—Sup. parietal (R)

NB. Ant. = Anterior, Sup. = superior, (L)/(R) = Left/Right, BA = Brodmann area).

**Table 6 ijms-23-07106-t006:** APOE4 status of those with clinical dementia or an AD pathology diagnosis in the ROSMAP RNA sequencing cohort.

APOE Status	No Dementia	Dementia	No AD Path	AD Path	Total
APOE4	58 (39%)	90 (61%)	29 (20%)	119 (80%)	148 (26%)
Non-carrier	285 (66%)	144 (34%)	198 (46%)	231 (54%)	429 (74%)
Total	343 (59%)	234 (41%)	227 (39%)	350 (61%)	577

NB—Percentages are to be read horizontally showing the diagnosis composition of APOE4 carriers and non-carriers. APOE4 = APOE 2/4, 3/4, 4/4; Non-carrier = APOE 2/2, 2/3, 3/3.

**Table 7 ijms-23-07106-t007:** Top 10 GO BP terms from enrichment analysis of the unique differentially expressed genes between dementia and no dementia groups in APOE4 carriers and non-carriers.

**DEGs Unique to APOE4 Carriers**			
**GO Term**	**Description**	***p*-Value**	**FDR**
GO:1901342	regulation of vasculature development	6.2 × 10^−6^	0.01
GO:0090130	tissue migration	6.7 × 10^−5^	0.03
GO:0001525	angiogenesis	1.5 × 10^−4^	0.04
GO:0034330	cell junction organization	2.4 × 10^−4^	0.05
GO:0001667	ameboidal-type cell migration	2.9 × 10^−4^	0.05
GO:0001570	vasculogenesis	8.0 × 10^−4^	0.11
GO:0040013	negative regulation of locomotion	1.9 × 10^−3^	0.20
GO:0048010	vascular endothelial growth factor receptor signalling pathway	1.9 × 10^−3^	0.20
GO:0002683	negative regulation of immune system process	2.1 × 10^−3^	0.20
GO:0007162	negative regulation of cell adhesion	2.3 × 10^−3^	0.20
**DEGs Unique to Non-Carriers**			
**GO Term**	**Description**	***p*-Value**	**FDR**
GO:0031348	negative regulation of defence response	6.7 × 10^−4^	0.32
GO:0061842	microtubule organizing centre localization	1.0 × 10^−3^	0.32
GO:0060491	regulation of cell projection assembly	1.1 × 10^−3^	0.32
GO:0046847	filopodium assembly	4.4 × 10^−3^	0.50
GO:0007050	cell cycle arrest	4.8 × 10^−3^	0.50
GO:1905475	regulation of protein localization to membrane	5.3 × 10^−3^	0.50
GO:0098927	vesicle-mediated transport between endosomal compartments	6.0 × 10^−3^	0.50
GO:0050727	regulation of inflammatory response	6.8 × 10^−3^	0.50
GO:0031032	actomyosin structure organization	6.8 × 10^−3^	0.50
GO:0016482	cytosolic transport	8.3 × 10^−3^	0.50

## Data Availability

Data was downloaded from Synapse website (https://www.synapse.org) following approval by Synapse Access and Compliance Team on 9 June 2021. Summary data is available in Appendix A. All code is available at https://github.com/binfnstats/ROSMAP_RNAseq.

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
