# Peer review of "Vascular Dysfunction Is Central to Alzheimer’s Disease Pathogenesis in APOE e4 Carriers"

_ijms, 2022, doi:10.3390/ijms23137106_

Round 1

Reviewer 1 Report

The association between APOE4 carrier status and increased AD risk has been well established, but the mechanisms involved remain uncertain. The authors address this key question by harnessing machine learning algorithms to look for correlates between APOE status, brain imaging and transcriptomic data utilising 3 different brain banks. From the UK Biobank (33,384 non-demented individuals), they establish baseline effects of APOE4 allele possession in white matter and grey matter regions, in under 65s and over 65s, respectively, particularly in posterior brain regions. In the ROSMAP cohort, average amyloid and tau pathology across 8 brain regions correlated with APOE4 allele dose, with neocortical areas exhibiting the highest fold-differences. It is also in this cohort that CAA pathology was identified as a high-ranking variable between APOE4 carriers and non-carriers. The vascular link was supported by a very significant (p<0.01) correlation of differential expressed genes related to vascular development in the demented group. In contrast to ROSMAP, APOE4 analysis on the Mount Sinai Brain Bank dataset gave limited differentially expression genes between the dementia (advanced AD) and non-dementia groups.

The paper is well-written, data clearly presented, and proper statistical tools (primarily Spearman correlations) were used to analyse the data.

Overall, the main conclusion from this study as suggested by the authors, that vascular dysfunction (in relation to CAA) may be significantly contributing to development of AD pathology in APOE4 carriers, is indeed supported by the data. 

However, one could comment that, although the authors’ methodological approach appears to be novel, the “APOE4-CAA-AD” link is not so original, and the authors should do better to cite and discuss previous work associating APOE4 carrier status with both AD and CAA. For instance, Shinohara et al. 2016 directly reported on the impact of APOE4 on CAA in AD, with APOE4 alleles being associated with more severe CAA in brain samples from pathologically confirmed AD cases. Meta-analysis by Rannikmae et al. 2014 also showed association of APOE4 with severe CAA. And, especially, in a close parallel to the conclusions of the present study, Thal et al. 2008 had concluded that hypoperfusion induced by capillary CAA, which is strongly associated with the APOE4 allele, should be considered as a mechanism by which amyloid-beta contributes to AD pathology.

Author Response

Dear Qian,

Thank you to the Reviewers for their comments on our manuscript entitled “Vascular dysfunction is central to Alzheimer’s disease pathogenesis in APOE e4 carriers”.  Please find below a point-by-point rebuttal and explanation of how our manuscript has been revised accordingly. We trust that these changes meet the Reviewers’ expectations.

The revised manuscript and a modified Supplementary table 3 has been uploaded as per instructions.

Please contact me if you require further information.

Yours sincerely

Greg (on behalf of my co-authors)

Reviewer 1

Overall, the main conclusion from this study as suggested by the authors, that vascular dysfunction (in relation to CAA) may be significantly contributing to development of AD pathology in APOE4 carriers, is indeed supported by the data. 

 Thank you to Reviewer 1 for their overall support of this study.

However, one could comment that, although the authors’ methodological approach appears to be novel, the “APOE4-CAA-AD” link is not so original, and the authors should do better to cite and discuss previous work associating APOE4 carrier status with both AD and CAA. For instance, Shinohara et al. 2016 directly reported on the impact of APOE4 on CAA in AD, with APOE4 alleles being associated with more severe CAA in brain samples from pathologically confirmed AD cases. Meta-analysis by Rannikmae et al. 2014 also showed association of APOE4 with severe CAA. And, especially, in a close parallel to the conclusions of the present study, Thal et al. 2008 had concluded that hypoperfusion induced by capillary CAA, which is strongly associated with the APOE4 allele, should be considered as a mechanism by which amyloid-beta contributes to AD pathology.

Thank you for these comments. As per suggestions by Reviewer 1 we have expanded Paragraph 4 of the Introduction to include a more thorough discourse on AD, CAA and APOE4. This includes citing the three interesting studies described by Reviewer 1. We have also cited these papers in our revised Discussion (Paragraph 3).

Reviewer 2 Report

The paper by McCorkingdale et al, describes studies where machine learning has been used to demonstrate brain structural changes linked to vascular dysfunction in Apoe4-carriers. The association between Apoe4 and vasculopathy was further back-up up by analysis on transcriptomic data from two dataset, where a link between Apoe4 and CAA as well as vascular related processes was verified. The study is of relevance given the increasing number of studies suggesting ApoE4-carriers as an AD subgroup with specific pathological events, something that should be accounted for in future diagnostic and treatments. 

The paper is well-written, methods are justified and limitations described and this reviewer has only few minor comments.

-In results (3.3) DEA was performed within diagnostic groups between APOE4 carriers and non-carriers in the ROSMAP-PFC dataset. A few DEGs were found, but these are not described anywhere. It would be interesting to take part in the content. 

- In results (3.3) there is a reference to table 8, but no table 8 can be found. 

-Add units into table one as it is easier to understand what is measured. 

Author Response

Reviewer 2

The paper is well-written, methods are justified and limitations described and this reviewer has only few minor comments.

 We thank Reviewer 2 for their overarching support of our manuscript.

-In results (3.3) DEA was performed within diagnostic groups between APOE4 carriers and non-carriers in the ROSMAP-PFC dataset. A few DEGs were found, but these are not described anywhere. It would be interesting to take part in the content. 

Thank you for pointing out this omission. In the revised Supplementary Table 3 (“ROSMAP”) we have now added an additional sheet called “Within diagnoses comparisons”. In this sheet we have listed the differentially expressed genes (DEG) of the four comparisons of APOE4 carriers/noncarriers and/or demented or non-demented. Please note that there were no DEGs between the APOE4 carriers and non-carrier non-demented individuals. Furthermore, in this additional sheet we have highlighted (in red) DEGs that were close to significance including LTF in the ‘NoAD’ (pathology)-APOE4 carrier/non-carrier comparison. We would also like to draw Reviewer 2’s attention to one of the DEGs from the ‘DEM_ APOE4 carrier/non-carrier ‘comparison, ADM. This encodes for adrenomedullin, which is a vasodilator expressed in cerebral endothelial cells. In the revised Discussion (paragraph 4, beginning “The molecular analysis…..”) we have included how ADM could regulate the blood-brain-barrier and thus could be a potential therapeutic target in AD and CAA.

- In results (3.3) there is a reference to table 8, but no table 8 can be found. 

Thank you for picking up on this error. This should have read Table 6 and has been corrected in the revised manuscript (Page 9, 2nd sentence).

-Add units into table one as it is easier to understand what is measured.

Thanks to Reviewer 2 for this suggestion. However ,there are no units here, as this table describes the number of individuals with each APOE genotype. We have modified the Table title to make this point more obvious to the reader.